# Programmable Data Plane Applications in 5G and Beyond Architectures: A Systematic Review

**DOI:** 10.3390/s23156955

**Published:** 2023-08-04

**Authors:** Jorge Andrés Brito, José Ignacio Moreno, Luis Miguel Contreras, Manuel Alvarez-Campana, Marta Blanco Caamaño

**Affiliations:** 1Departamento de Ingeniería de Sistemas Telemáticos, ETSI de Telecomunicación, Universidad Politécnica de Madrid, 28040 Madrid, Spain; joseignacio.moreno@upm.es (J.I.M.); manuel.alvarez-campana@upm.es (M.A.-C.); 2Telefónica I+D, 28010 Madrid, Spain; luismiguel.contrerasmurillo@telefonica.com (L.M.C.); marta.blancocaamano@telefonica.com (M.B.C.)

**Keywords:** programmable data plane, 5G and beyond, architectures, programmable devices, P4, SDN

## Abstract

The rapid evolution of 5G and beyond technologies has sparked an unprecedented surge in the need for networking infrastructure that can deliver high speed, minimal latency, and remarkable flexibility. The programmable data plane, which enables the dynamic reconfiguration of network functions and protocols, is becoming increasingly important in meeting these requirements. This paper provides an overview of the current state of the art in programmable data planes implemented in 5G and beyond architectures. It proposes a classification of the reviewed studies based on system architecture and specific use cases. Furthermore, the article surveys the primary applications of programmable devices in emerging telecommunication networks, such as tunneling and forwarding, network slicing, cybersecurity, and in-band telemetry. Finally, this publication summarizes the open research challenges and future directions. In addition to offering a comprehensive review of programmable data plane applications in telecommunication networks, this article aims to guide further research in this promising field for network operators and researchers alike.

## 1. Introduction

Programmable data planes have emerged as a critical component of modern networking architectures, enabling greater flexibility, faster processing, and more efficient use of network resources. Following the principles of software-defined networking (SDN) [1] in programmable data planes, the control plane and data plane are separated, allowing for more granular control over network traffic and enabling administrators to configure and adapt networks to changing requirements and traffic patterns.

To enable programming of the data plane, a variety of programming languages have been developed throughout the years, each offering distinct attributes and functionalities, e.g., Domino [2], Lucid [3], NetKAT [4], OpenState [5], and P4 [6]. However, P4 (Programming Protocol-Independent Packet Processors) is the most widely used language in the state of the art. Furthermore, this language enables the creation of custom packet processing pipelines, allowing for the implementation of advanced features and services, such as traffic monitoring, load balancing, function offloading, and security filtering.

The evolution of communication architectures, including the emergence of 5G and beyond [7], has created opportunities for SDN and programmable data planes to show their potential as key enabler technologies. These architectures handle vast amounts of users and their data, requiring rapid and effective processing to cater to a diverse array of applications and services [8]. SDN, in this context, offers architectural flexibility and scalability to address the complexities of managing a large number of connected devices. By utilizing centralized controllers, SDN simplifies management and automates tasks such as provisioning, configuration, and troubleshooting within telecommunication network infrastructure. Its programmable and dynamic nature enables operators to swiftly reconfigure the network and allocate resources based on specific use case requirements [9]. Complementarily, programmable data planes can be used to support low-latency applications, such as autonomous vehicles or industrial automation, by providing fast and efficient pipelines. They could also be employed to provide higher throughput in user plane functions and help reach new levels of mobile broadband connectivity.

The integration of programmable data planes in 5G and beyond architectures presents both advantages and potential impacts, but it also introduces implementation challenges. By offloading specific network function tasks to programmable devices, the overall network performance can be significantly optimized, thereby fulfilling the stringent requirements and key performance indicators (KPIs) of 5G and upcoming technologies [10,11]. Moreover, the inherent customization capability of these devices enhance support for network slicing and multi-access edge computing, essential features in novel telecommunication systems. However, the deployment of programmable data planes poses challenges in efficiently managing a vast number of users, ensuring seamless communication between different vendor network devices, and overcoming the limitations of data plane programming languages for novel use cases. Despite these hurdles, programmable data planes have the potential to enable mobile networks to operate at optimal levels, thus enhancing user experience. Furthermore, they accelerate innovation cycles and enable experimentation, fostering the development of more advanced and transformative telecommunication networks.

In the following subsections, the contribution of this review will be presented in detail along with its objectives and distinctive features compared to other reviews.

### 1.1. Contributions

This article presents a systematic review of programmable data plane applications in 5G and beyond networks along with relevant commercial implementations. This work targets applications that use P4, which is the most widely used language in both academia and industry. The contributions of this survey can be summarized as follows:Providing the first review that exclusively focuses on programmable data plane implementations on 5G and beyond architectures.Offering a comprehensive and up-to-date review of research work on these novel technologies.Proposing a classification of programmable data plane implementations based on 5G and beyond architectural components, as well as their use cases, categorizing 59 research papers and 2 commercial solutions.Identifying open challenges and future research directions in the field.Providing information about the implemented device and the code availability for each of the surveyed articles.

### 1.2. Related Reviews

The literature covers other surveys that examine programmable data plane applications in telecommunication networks, including those for 4G and 5G. Kfoury et al. [12] present a comprehensive review of programmable data plane switches and P4 language, encompassing various applications, including telecommunication services. The authors analyze each study, compare their results, and discuss limitations. However, the paper has a limited number of articles related to 5G and its associated technologies and does not include recent studies as it was published in 2021.

Liatifis et al. [13] offer an overview of P4 implementations in various fields, such as next-generation mobile networks. Nevertheless, this survey compiles a limited selection of studies and does not extensively analyze the detailed findings reported within them. Although the authors propose future research directions, there are no specific prospects for telecommunication networks.

Hauser et al. [14] provide a tutorial on data plane programming and an extensive survey of P4 and applied research, including cellular networks. The article includes an analysis of the results and potential new research directions. However, it lacks an architectural classification for 4G/5G applications and does not include recent studies in the field.

Kianpisheh et al. [15] present a survey of programmable data planes, recognizing their role as an enabler for diverse in-network computing (INC) applications. The article provides a technical definition of INC and introduces a thorough classification of its uses, including 4G/5G/6G technologies. In this context, the authors analyze various implementations and conduct comparative assessments of their results. However, this study does not include several articles with relevant applications and does not provide information about code availability.

## 2. Methodology

This section presents the methodology and criteria employed to search, select, and filter the articles included in the review. Additionally, it provides details about the analysis performed on the extracted information.

### 2.1. Article Search and Selection

To conduct the review, an extensive search was performed using reliable sources indexed by digital object identifier (DOI). This was accomplished using the following digital library platforms:IEEE Xplore Digital LibraryGoogle ScholarScopusWeb of Science

This search was based on programmable data plane terminology and 5G and beyond technology. Specific keywords within articles were used, such as Programmable data plane, PDP, P4, SDN, 5G, 6G, UPF, and network slicing. Finally, Boolean operators such as “AND” and “OR” were utilized along with the keywords to refine the search and increase its scope.

### 2.2. Inclusion and Exclusion Criteria

In order to refine the results of the keyword search, a set of inclusion and exclusion criteria was applied to select studies relevant to the objectives of this review. Initially, only articles that described programmable data plane implementations within 5G and beyond architectures were included. Specifically, studies that utilized the P4 programming language were considered, as it is widely recognized as the de facto standard in the current state of the art. Subsequently, a second criterion was applied to limit the selection to studies published between 2018 and 2022 as well early work from 2023. The reason for selecting this time frame is that it corresponds to the period during which the vast majority of research on programmable data planes in telecommunication networks was conducted. Prior to 2018, publications on this topic were scarce and primarily focused on pre-5G technology. It is worth noting that certain 4G LTE applications were considered within this review given their compatibility with 5G and beyond systems. 

The primary search yielded 224 research articles, which were further reduced to a final set of 61 articles that met the aforementioned criteria and were deemed relevant to the objectives of this review.

### 2.3. Data Extraction and Analysis

A thorough inspection of the selected articles was conducted, carefully extracting detailed information about the various applications under consideration. This included technical evidence of the implementations as well as the reported results. These data were then classified and synthesized to allow its proper dissemination in the review.

The analysis of the extracted data aimed to identify gaps in the existing research and to highlight potential new research opportunities for programmable data planes in emerging telecommunication network architectures.

## 3. Background in Programmable Data Plane and 5G and Beyond

This section provides an overview of programmable data planes and their enabling technologies, such as SDN, PISA architecture, and P4 programming language. It also covers the programmable devices that support these features. Additionally, a comprehensive review of 5G and beyond architectures is provided to contextualize the implementations described in the survey. Finally, relevant paradigms that play a key role in the analyzed use cases are explained, including network function virtualization (NFV), network slicing (NS), and multi-access edge computing (MEC).

### 3.1. Software Defined Networking, Programmable Data Plane, and Devices

#### 3.1.1. Software-Defined Networking

Software-defined networking (SDN) is an emerging networking paradigm that enables network administrators to oversee and govern the network infrastructure using software applications, instead of depending solely on conventional networking hardware [1]. SDN works by decoupling the control plane from the data plane, allowing the network to be centrally managed and dynamically configured. This is accomplished through the use of a centralized controller that communicates with switches and routers in the network using a standardized protocol, such as OpenFlow [16]. The principles of SDN can be summarized as follows:Separation of control plane and data plane: The control plane, responsible for governing forwarding behavior, is separated from the data plane, which performs the actual traffic forwarding based on instructions from the control plane. This decoupling enhances the network architecture’s flexibility and scalability, allowing for more efficient management.Centralized Control: The so-called SDN controller is responsible for handling control logic, being a high-level software program that can run on commodity servers.Programmability: The network can be programmed using software applications that run on top of the SDN controller, enabling dynamic configuration and automation of network functions.Virtualization: The network can be virtualized, enabling multiple virtual networks to run on a single physical network infrastructure.Open Standards: SDN uses open standards and protocols, enabling interoperability between different vendors’ products and facilitating new developments.

By adopting SDN, network administrators can create a more agile, flexible, and scalable network infrastructure providing better control and visibility over network traffic.

Figure 1 illustrates a general architecture of SDN. The network is managed through network applications, which can include load balancing, firewall management, and network monitoring. The controller platform integrates the SDN controller, which manages network devices and enforces network policies. Communication between the controller and network devices is facilitated through protocols such as OpenFlow or vendor-specific alternatives. Network devices, such as switches and routers, are responsible for forwarding data packets between network nodes, constituting the data forwarding elements. To define the data plane behavior of SDN devices, P4 can be employed, specifying packet parsing, processing, and forwarding operations.

#### 3.1.2. Data Plane Programmability

Data plane programmability is a concept that refers to the ability to customize and manipulate the forwarding behavior of network devices, by programming the data plane hardware or software directly. The data plane is responsible for the actual transfer of data packets across the network and is typically implemented in high-speed packet processing devices.

Traditionally, the data plane has been fixed and determined by the firmware of the network devices, with limited ability for customization or dynamic adaptation to changing network conditions. However, with the rise of SDN, data plane programmability has become increasingly important as a means of enabling flexible and dynamic network architectures that can respond to changing traffic patterns, application requirements, and security threats. A few examples of new use cases are in-band network telemetry (INT), active queue management (AQM), time-sensitive networking (TSN), and traffic offloading.

#### 3.1.3. PISA (Protocol Independent Switch Architecture)

The Protocol-Independent Switch Architecture (PISA) is a programmable switch model that enables granular control over packet processing [17]. As shown in Figure 2, PISA comprises a programmable parser, a programmable match-action pipeline, and a programmable deparser that work together to deal with incoming packets [18].

The programmable parser is responsible for extracting headers from incoming packets and parsing them based on custom or standard protocol. It can be represented as a state machine, enabling granular control over packet processing. The programmable match-action pipeline is the core component of PISA and executes operations over the packet headers. It is based on the concept of a programmable table and consists of multiple stages that process headers using match-action tables (MATs) and forward them to the next stage. This allows for simultaneous lookups and actions across multiple memory blocks and arithmetic logic units (ALUs). The programmable deparser is responsible for reassembling the packet headers and serializing them for transmission. It receives the processed headers from the pipeline stages and combines them to reconstruct the original packet, ensuring that the packet is correctly formatted and ready to be transmitted.

#### 3.1.4. P4

Programming Protocol-Independent Packet Processors, P4 [6], is a high-level domain-specific programming language designed for programming network data planes using the PISA processing pipeline. This enables the development of adaptable, customizable devices that can be precisely adapted to meet the demands of specific applications and network configurations. The main principles of P4 are:Programmability: Network operators define how packets are parsed and processed in a way that is both adaptable and extensible.Protocol Independence: Packets are processed independently of the underlying protocols or technologies used in the network.Match-Action Pipeline: A match-action pipeline model (i.e., PISA) is used to process packets. In this model, incoming packets are matched against a set of rules that define how they should be processed.Target-Independent: Code can be compiled to run on various network devices, such as switches, routers, and programmable network interface cards (NICs) regardless of the specific target.

The P4 Language Consortium has undertaken the development and standardization of P4 as a programming language [19]. It has undergone several revisions since its initial specification in 2014, with P4_14_ being the first standardized version [20]. Introduced in 2016, the P4_16_ specification [21] builds upon its predecessors and offers a broader range of capabilities. It extends the applicability of P4 to cover a diverse range of targets, including application-specific integrated circuits (ASICs), field programmable gate arrays (FPGAs), and network interface cards (NICs).

#### 3.1.5. Programmable Devices

The programmable data plane functionalities can be implemented in a variety of network devices. From which the following categories can be identified:Programmable switches: Similar to traditional network switches but with programmability capabilities. The match-action pipeline is the fundamental abstraction for the functionality of a programmable switch. Thus, these devices use the PISA architecture in their design. They can also be classified into hardware and software switches. The formers are based on ASICs (application-specific integrated circuits) such as Intel Tofino (e.g., EdgeCore Wedge 100BF-32X from Edgecore Networks (Hsinchu, Taiwan) [22], Inventec D10056 from Inventec Corporation (Taipei, Taiwan) [23], and Netberg Aurora 610 from Netberg (Taoyuan, Taiwan) [24]). On the other hand, software switches are programs for forwarding packets that operate on regular CPUs (e.g., bmv2, p4c-behavioral and T4P4S). The bmv2 switch can reach 1 Gbps [25], while the latest Tofino ASIC (Intel Tofino 2) offers rates of 12.8 Tbps [26].FPGA boards: Development boards that have as the main component an FPGA (Field Programmable Gate Array). FPGAs are semiconductor elements that are reconfigurable and can be programmed to implement custom hardware functionality. These devices also incorporate SFP and PCIe interfaces for high-density networking. NetFPGA PLUS [27] and NetFPGA SUME [28] are examples of these boards. Based on Xilinx FPGAs, they can achieve a throughput up to 100 Gbps.Smart NICs: Programmable NICs that offload network processing tasks from the host CPU to a dedicated hardware accelerator (i.e., network processing unit). Well-known smart NICs include Netronome Agilio CX series [29]. The latter being able to perform 100 Gbps at line rate.

Even though ASIC-based switches offer the highest throughput, they are less flexible in terms of programmability (i.e., processing of external functions). FPGA boards and Smart NICs offer further programmability but with more modest data rates. Finally, software switches have the highest degree of flexibility at the cost of lower throughput.

### 3.2. 5G and Beyond Technology and Architecture

#### 3.2.1. 5G Technology

5G technology has been developed by the 3rd Generation Partnership Project (3GPP), which has set out to define and finalize its specification over Releases 15, 16, 17, and 18 [7]. 5G aims to deliver enhanced data rates exceeding the capabilities of existing 4G LTE networks. Moreover, it offers wider network coverage, more reliable connections, reduced latency times, lower power consumption, and improved scalability [30]. These capabilities allow the implementation of new applications, also known as verticals, such as: eHealth, autonomous vehicles (V2X), virtual/augmented reality (VR/AR), smart cities, smart homes, and Industry 4.0. Accordingly, the IMT-2020 (International Mobile Telecommunications—2020), issued by the ITU-T [8], has defined three categories to classify 5G applications, as can be observed in Figure 3:

Enhanced mobile broadband (eMBB): Designed to deliver high data rates (up to >> 1 Gbps), allowing users to experience new levels of mobile broadband connectivity. It supports services like virtual reality, ultra-high-definition video streaming, or immersive gaming.Ultra-reliable and low-latency communications (uRLLC): Enables devices to communicate with each other in “real time”. This category of service is applied in scenarios where data loss must be avoided, low latency is crucial, and a high level of reliability is required. Applications such as V2X, distribution automation in a smart grid, or remote medical surgery are supported.Massive machine type communications (mMTC): Enables massive numbers of devices to be connected. Usually, these devices transmit relatively low volumes of non-delay-sensitive data. Backed services include IoT applications: Smart cities, smart homes, or some industrial IoT scenarios.

The aforementioned usage scenarios are associated to a number of KPIs defined for IMT-2020 in [10]. In this manner, the KPIs have minimum technical demands to fulfill in order to be 5G complaint, e.g., user plane latency of 4 ms for eMBB and 1 ms for uRLLC. Table 1 reports a complete list of these requirements.

#### 3.2.2. 5G System Architecture

In the first stage of Release 15, 3GPP has defined the complete architecture of a 5G system (5GS) [31]. It describes a set of characteristics and specifications necessary for the deployment of an operational infrastructure for mobile networks based on a 5GS. During the second stage, the mechanisms for data and service connectivity, as well as the general 5G architecture, were established. Figure 4 illustrates a simplified 5GS architecture based on a service-based representation [32]. In this description, every network function (NF) (e.g., NSSF) provides a service which can be consumed by any other NF (e.g., AMF). NFs interact with each other using either dedicated protocol point-to-point interfaces (e.g., N1) or through API-based interfaces (e.g., Nnssf). As can be observed, there is a separation of the control plane and the user plane.

#### 3.2.3. Network Functions, Entities, and Subsystems

There are a vast number of NFs, entities and subsystems included in the 5GS architecture. However, for the sake of brevity, this subsection will only describe the components that are the most important to the implementations included in the review.

5G residential gateway (5G-RG): Device that enables residential or small business fixed users to connect to a 5G network and then to a DN such as the Internet. This acts as a bridge between 5GC and UE.Access and mobility management function (AMF): Control plane NF within the 5G Core (5GC). The UEs transmit all connection- and session-related data to the AMF, which is in charge of connection and mobility management duties. Other key features are cyphering and integrity protection, providing the user equipment (UE) with a temporary ID, subscriber authentication, support for location services (cell sites or tracking area), and help in lawful interception.Access gateway function (AGF): NF that enables fixed users to receive services from the same 5GC that serves mobile subscribers. Its key functions include handling signaling associated with QoS and PDU sessions as well as marking user plane packets in uplink connections.Authentication server function (AUSF): Manages UE authentication of a 3GPP or non-3GPP access.Data network (DN): In addition to IP-based data networks (e.g., the Internet), it refers to any other structured data network (e.g., IoT data with low overhead).Network repository function (NRF): Works as a central repository for all NFs. Allows NFs to be registered and recognized.Network slice selection function (NSSF): Aids in selecting the network slice instance that will support a given device. The concept of network slicing will be further described in the following subsection.Next-generation radio access network (NG-RAN): Constitutes the 5G radio access for the UE. The main component of this subsystem is the 5G Node B (gNB), i.e., the 5G New Radio (5G-NR) base station. It can be separated into two modules: central unit (CU) and distributed unit (DU). This architecture features connections among CU, DU, and 5GC. The CU handles upper layers and can be deployed as a hardware device or as cloud-based software. While DUs are placed at cell sites and manage time-sensitive processes. There are architectural variations for NG-RAN (e.g., C-RAN), further details can be found in [33].Policy control function (PCF): Establishes unified policy rules for control NFs like mobility, roaming, and slicing.Session management function (SMF): Another control NF, it is responsible of the session management, i.e., creation, update, and termination of the PDU (protocol data unit) session. Further capabilities include IP address allocation for UE, selection and control of user plane function (UPF), and liaison with the policy control function (PCF) for policy and QoS enforcement.Unified data management (UDM): Stores subscriber data and user profiles.User equipment (UE): Any end-user device that is able to connect to a 5G network, e.g., a mobile phone, an IoT sensor node, or a vehicle.User plane function (UPF): Handles UE data traffic by routing and forwarding packets. It also acts as an interconnection point between NG-RAN and DN, providing GPRS tunnelling protocol (GTP) encapsulation and decapsulation. Other important functionalities are acting as an anchor for RAN mobility, applying service data flow (SDF) filtering, implementing per-flow QoS ID, and reporting of traffic usage for billing.

Given the line rate characteristic of data plane programmable devices, it makes them ideal candidates for handling 5G user traffic. Thus, a common implementation involves offloading UPF forwarding services to ASIC-based switches. Other uses include employing programmable devices within 5G entities such as gNB or even at the edge of the network (between NG-RAN and 5GC).

#### 3.2.4. Beyond 5G Prospectives

Research beyond 5G technology has already started [34]. 6G is the next generation of communication networks that will succeed the current 5G networks. 6G is expected to provide faster speeds, lower latency, enhance mobility, and more reliable connections than 5G [35]. It is also expected to enable new applications and use cases that are not currently viable with existing technology. Some of the potential use cases for 6G include advanced virtual and augmented reality (truly immersive XR), holographic communication (high-fidelity mobile hologram), and digital replicas (digital twin) [36]. These applications will shape new requirements and KPIs for 6G.

Figure 5 shows the requirements for 6G. As can be observed, three of them are similar to 5G. However, there are three new categories [11]. Precision and accuracy are associated with sensing and localization capabilities, respectively. In this context, one way of measuring sensing precision is using missed detection (MD) and false alarm (FA) parameters. There is also adaptability response time, which is related to network automation and the quantity of manual intervention that it needs. Finally, end devices are expected to have intuitive interfaces, e.g., sensible to gestures, as well as to consume from low power to no power at all. Furthermore, some key enabling technologies that are expected to be part of 6G research include artificial intelligence/machine learning, terahertz technologies, spectrum sharing, and new network architecture paradigms such as converged RAN-Core and subnetworks [37].

The exploration of potential applications and capabilities of 6G technology is an ongoing endeavor. It is anticipated to bring forth substantial advancements in communication networks and facilitate novel and inventive use cases spanning various industries.

### 3.3. Network Function Virtualization, Network Slicing, and Multi-Access Edge Computing

#### 3.3.1. Network Function Virtualization (NFV)

Network function virtualization (NFV) [38] is an architectural approach for designing and deploying network services by virtualizing the functionality of traditional dedicated hardware, such as routers, switches, firewalls, and load balancers, into software running on standard servers, storage, and networking equipment.

This architecture has been standardized by ETSI (European Telecommunications Standards Institute) [39] defining functional building blocks and interfaces to manage virtualized network services. 5G systems incorporate virtualization technology to enable the implementation of various network services, including 5G core network functions, such as the AMF as virtual network functions. This means that 5G network functions can be executed on standard servers, which can be located in centralized data centers or cloud-based infrastructures.

The NFV architecture presents a structural basis for constructing adaptable and versatile networks through the virtualization of network functions and their separation from the underlying hardware. This permits network operators to enhance scalability and optimize operational expenses by leveraging a shared infrastructure to cater to multiple network services effectively.

#### 3.3.2. Network Slicing (NS)

Network slicing (NS) [40] is a technique employed in data networks that enables the establishment of multiple virtual networks, known as slices, atop a single physical network infrastructure. These slices are distinct and self-contained network instances capable of being tailored to satisfy the specific demands of an application or service.

NS finds significant utility in 5G networks [41], in which slices are utilized to deliver differentiated services for diverse applications, such as IoT (mMTC), autonomous vehicles (uRLLC), and audio/video streaming (eMBB). Each of these applications possesses unique network requirements pertaining to bandwidth, latency, and reliability. This facilitates the provision of individual virtual networks for each application, accompanied by their own QoS assurances.

#### 3.3.3. Multi-Access Edge Computing (MEC)

Multi-access edge computing (MEC) [42] is a distributed computing paradigm aimed at extending the capabilities of cloud computing in proximity to end-user devices. By leveraging an infrastructure that enables the execution of applications and services at the network’s edge, MEC reduces reliance on centralized cloud computing resources. These resources can be strategically located in diverse areas such as base stations, access points, and edge routers. This approach holds the potential to decrease network latency, optimize network bandwidth utilization, and facilitate the development of innovative applications and services that demand real-time data processing and low-latency response times. Within the realm of 5G, there exist several deployment scenarios wherein MEC can seamlessly integrate with 5GS [43]. Figure 6 showcases four plausible physical implementations of a 5G + MEC network.

## 4. Findings and Discussion

This section presents an analysis of the contributions made by the surveyed papers and a discussion of their results. Firstly, the criteria used to classify the proposed articles are provided, based on the role of programmable data plane devices within the 5G and beyond architecture. Subsequently, the implementation of each selected work is explained and compared, including their reported results. Finally, a summary of the findings is presented along with the insights gained from the research.

### 4.1. Classification of the Reviewed Papers

Programmable data planes have demonstrated significant utility in novel telecommunication systems, as highlighted in previous sections. Numerous studies have explored the potential of these applications, making it essential for network operators and researchers to have a systematic review of these studies, placed within a system architecture setting. This approach aids in the identification of critical subsystems and network functions that incorporate programmable devices. To address this need, a classification of articles is proposed centered on 5G and beyond architecture and its use cases. Table 2 presents the position of each reviewed work within the system architecture, its associated use case category, and the deployed programmable data plane device.

### 4.2. Reviewed Literature Statistics

Upon examining the 61 articles included in this review, relevant statistics were obtained. Figure 7 presents the distribution of papers per year from 2018 through 2022 as well as early work from 2023.

The distribution of programmable devices used in the reviewed articles is presented in Figure 8. The results show that a significant portion of the implementations were done using software switches (40%). These switches are typically used for prototyping and proof of concept purposes. Hardware devices, including switches, FPGA boards, and smart NICs, accounted for a combined usage of 55%. This indicates that the majority of research work is geared towards deployment in real-world networks. It should be noted that 4.9% of the papers did not provide any information regarding the device used.

Figure 9 shows the distribution of papers based on the architecture section where the applications have been deployed. It is evident that the majority of solutions, representing 51%, have been implemented in the edge-to-core section of the architecture. This indicates that most of the research found in this review is focused on the transport network. In addition, Figure 10 illustrates the distribution of articles based on their specific use case implementations, revealing that tunneling and forwarding, alongside network slicing, emerge as the most frequently adopted use cases. Moreover, the survey found that 16 articles have made their used code available on the GitHub platform. Further details can be found in Table A1 from Appendix A of this paper.

### 4.3. Characteristics of the Reviewed work

Aghdai et al. [44] propose a transparent edge gateway (EGW) for MEC in LTE or 5G networks. This solution is implemented on P4 smart NICs, enabling content delivery at the edge of the transport network by parsing the inner IP headers of GTP-U messages. In their subsequent work [45], the authors incorporated mobility support for the handover process, aiming to minimize the number of application state migrations.

Shen et al. [46] introduce a GTP engine for MEC in 5G networks, which offloads the encapsulation and decapsulation of the tunneling protocol to a P4 FPGA board.

Lee et al. [47] present a P4 switch that implements stateless translation functions for both GTP and SRv6 (Segment Routing IPv6) protocols. This approach enables the coexistence of both protocols within a 5G network, facilitating a gradual transition towards full adoption of SRv6.

Singh et al. [48] implement evolved packet gateway (EPG) offloading by employing P4 switches to perform functions such as GTP and VXLAN (virtual extensible local area network) encapsulation/decapsulation, IP routing, and stateless firewall. In their follow-up work [49], the authors present a hybrid pipeline design for 5G gNB and UPF. P4 smart NICs and switches were used to handle most of the packet processing, while unsupported functions such as ARQ (automatic repeat request) and cryptography were performed using DPDK (data plane development kit).

Vörös et al. [50] introduce a hybrid approach to packet processing in gNB that employs a P4 switch for most of the workload and outsources additional tasks, such as ARQ and ciphering, to DPDK external services. This approach allows for a more efficient utilization of hardware resources and better scalability of the system.

Ricart-Sanchez et al. [51] present a P4-based solution to improve the performance of the edge-to-core network data path in 5G multi-tenant environments. The proposed solution leverages P4 FPGA boards to handle encapsulation protocols such as VXLAN, GTP, and GTP over VXLAN, enabling efficient traffic routing and forwarding. In their subsequent works [88,89], the authors introduce a firewall system for 5G multi-tenant scenarios that supports traffic detection, differentiation, and selective blocking in the backhaul network. The firewall rules are stored in the TCAM (ternary-content-addressable memory) table of the P4 FPGA boards. Moreover, in [72,73], the authors design and implement a network slicing framework for the edge-to-core network segment that allows for the creation of different slices based on a 6-tuple consisting of user source and destination IPs, user source and destination ports, differentiated services code point (DSCP), and GTP tunnel ID. The framework is deployed in a smart grid self-healing automatic reconfiguration use case in [74], i.e., uRLLC traffic.

Lin et al. [52] leverage P4 switches to implement a network slicer and a UPF for handling different types of traffic in a 5G testbed. Specifically, they use P4 switches to implement the transport network slicing functionality as well as the data plane functions required for supporting mMTC, eMBB, and CIoT traffic.

NIKSS [53] is a software switch implemented in P4. It features a PSA (portable switch architecture) eBPF compiler that translates P4 programs into executable code. This device has been programmed to function as a 5G UPF for evaluation purposes, demonstrating its capability to handle a range of protocols, including IP, UDP, and GTP-U.

MacDavid et al. [54] implemented two 5G UPFs. The first is a model UPF implemented as a P4 software switch, with the PFCP interface defined as a series of match-action tables based on packet metadata. This approach provides developers with a useful starting point for creating full-fledged UPF implementations for specific hardware targets. The second UPF is designed to run on hardware switches, maximizing bandwidth and minimizing latency. It leverages microservices to provide additional functionality, such as buffering traffic for idle mobile devices.

Alfredsson et al. [55] propose the design of a 5G multi-access proxy’s user plane based on the MP-DCCP (multi-path datagram congestion control protocol) protocol, which extends DCCP to support multipath communication. To implement this design, the authors use a P4 smart NIC.

Bose et al. [56] implement 5G UPF prototypes, one of which utilizes a P4 smart NIC to offload both data plane processing (e.g., GTP encapsulation/decapsulation and oversubscribed session queuing) and control plane signaling (e.g., control plane packet processing and data plane rule installation). In a follow-up work [57], the authors propose AccelUPF, a high-performance 5G UPF that offloads user plane functionality to a programmable switch, achieving significant acceleration in data plane processing. It also offloads PFCP (packet forwarding control protocol) message processing from the control plane, dividing it between the hardware and software components to optimize performance. The fast path on hardware is assigned to the more common and simpler patterns of PFCP messages, while the software handles the more complex and infrequent messages.

CeUPF [58] adopts a hybrid architecture that combines software and hardware elements to optimize the performance of user plane functions. Specifically, some actions, such as traffic steering and datagram forwarding, are offloaded to programmable hardware. While other messages that the hardware cannot process are directed back to the software user plane. This offloading is achieved using a P4-based hardware switch and a smart NIC.

Rischke et al. [59] conducted a comparative analysis of XDP, DPDK, and P4 as processing acceleration technologies in uRLLC scenarios. In their study, they implemented UPF surrogates to handle GTP processes for each technology. Specifically, they used a P4 FPGA board as the device under test to evaluate the performance of the P4-based UPF.

Fernando et al. [60] develop a 5G-MEC testbed for the purpose of gathering data related to cybersecurity. To achieve high throughput in this architecture, P4 switches are utilized specifically for the handling of UDP traffic.

Jain et al. [61] utilize a P4 Smart NIC to implement the UPF function for 5G and beyond networks. The device performs forwarding and tunneling, while more complex functions such as buffering and flow processing are assigned to a host-based UPF.

Gramaglia et al. [62] present an implementation of SRv6 as a transport protocol for 5G network slicing utilizing P4 switches. The proposed solution is evaluated in terms of performance against the widely used GTP protocol.

BRAINE [63] proposes a MEC solution for 5G. This framework deploys multiple P4 switches to perform INT, user plane offloading, and DoS attack detection.

Kong et al. [64] implement a MEC router based on a P4 switch to provide ultra-low latency services in a distributed computing environment. The router performs GTP match and encapsulation/decapsulation actions, which minimize the load on host CPU cores.

Synergy [65] is a high-performance UPF on P4 smart NICs with monitoring capabilities for user session data prediction and handover optimization. Efficient buffering of data packets during handover and paging events is accomplished by employing a two-level flow-state access mechanism, which results in low latency for both control and data planes while maintaining high packet forwarding throughput. The prediction of handover events is achieved through the utilization of a recurrent neural network model.

Velox [66,67] is a network of P4 switches designed to interconnect 5G RAN and core networks for industrial scenarios. It enables switches to process cellular control (NGAP) and cellular data packets (GTP) while introducing the concept of intra-cellular optimization to reduce latency between two devices on the same network. Additionally, active monitoring and security capabilities are included in the data plane pipeline.

Paolucci et al. [68] present an implementation of a UPF offloaded to a P4 switch. The demonstration includes GTP-U encapsulation/decapsulation functions, automatic forwarding and steering functions, and configurable monitoring of selected GTP flows’ performance, such as the online latency experienced at a node. Additionally, in [89], the same authors introduce use cases that showcase the potential of programmable data planes in 5G SDN networks. P4 switches are utilized to provide advanced functionalities such as traffic engineering, cybersecurity, multi-tenancy, 5G offloading, and telemetry.

Kundel et al. [69] evaluate various UPF implementations in an end-to-end 5G standalone test network, including a P4 switch based UPF.

The Kaloom 5G UPF [70] employs an Intel Tofino ASIC and offloads QoS processing (i.e., bit rate policing) and GTP processing to the programmable hardware. Additionally, it supports network slicing and SRv6.

Metaswitch Fusion Core [71] is a private 5G Core designed for MEC. It features a UPF that carries out packet classification, routing, and forwarding tasks. The platform includes a cross-compiler that facilitates the use of the P4 programming language for defining UPF pipelines.

Cunha et al. [75] and Chang et al. [76,77] present a solution that aims to ensure performance isolation for network slicing, specifically with regards to bandwidth and delay guarantees, in order to support three Industry 4.0 case scenarios, such as digital twin, telemetry, and remote support. The proposed implementation integrates P4- and OpenFlow-based switches at the transport network data plane. The former deploy packet marking and meter coloring actions to provide the necessary network slicing functionality.

Chiu et al. [78] propose a comprehensive framework for achieving end-to-end network slicing in 5G networks. In their approach, P4 switches are utilized to enable transport network slicing, allowing for the enforcement of slice identification and bandwidth control in accordance with the QoS requirements of each slice.

Wang et al. [79] propose a network slicing framework for an eHealth use case in the 5G context. The framework employs P4 FPGA boards to perform traffic parsing and classification, as well as QoS control, for video transmission. To enable QoS control, an API is utilized to specify the priority of the network traffic processed by the board.

FestNet [80] is a sliced transport network, which utilizes P4 switches to implement a virtualized programmable data plane (vPDP) and a two-layer design. This enables the provision of network slicing and live slice mobility functionalities.

FSA [81,82] provides dynamic network slicing by utilizing the wireless schedule to identify the slice for each fronthaul packet. This architecture uses P4 switches and enables packet prioritization.

Yan et al. [83] propose an optical 5G inter-data center architecture that utilizes a P4 FPGA board as server-edge processor. The architecture facilitates network slicing and inter-data center communication.

P4-TINS [84] is a solution that provides bandwidth guarantee and management for network slices. The solution adopts a two-level priority queue framework in which a meter serving each slice receives all traffic belonging to that slice and dispatches packets to high- and low-priority queues. This ensures that there is no interference between slices.

AHAB [85] is a hierarchical per-user bandwidth allocator designed for network slicing. This solution operates directly in the data plane using a P4 switch which dynamically adjusts the user bandwidth limit for each slice in real-time. It adopts a maximum–minimum fairness approach that considers the bandwidth demand of all users across all slices, thereby avoiding the need to store per-user state in switch memory.

Turkovic et al. [86] propose a P4 switch-based network slicing framework that can handle time-sensitive tasks such as overload and underload detection, rerouting, and state transfer. A custom slice management protocol is implemented using a SM header to enable efficient slice management.

Lin et al. [87] propose a content permutation algorithm for handling IoT traffic in 5G networks. Their approach involves implementing the algorithm in P4 switches, where packet payloads are split into code words and shuffled according to a secret cipher generated at an SDN controller.

Wen et al. [91] propose a virtual testbed for 5G security experimentation, including a P4 switch for prototyping defense mechanisms and developing traffic rules. The defense mechanism uses a modified countmin sketch data structure to detect UEs exceeding a threshold, limiting the maximum bit rate with a TrTCM meter.

FrameRTP4 [92] is a framework that aims to deliver real-time attack detection and mitigation mechanisms in network slicing. For this, it provides a customizable P4 program that includes a service function chain to enable the lifecycle management of slices. Additionally, the P4 program deploys a monitoring system, namely SFCMon, that uses bloom filters and sketches to support a mechanism to track network flows.

Dreibholz et al. [93] describe a 4G/5G testbed where P4 switches are used to perform in-band network telemetry. The telemetry data is sent to a collector that gathers information about traffic flows and switch queue status. This approach enables end-to-end performance testing to enhance quality of experience (QoE) for end-users.

Scano et al. [94] propose a P4-based INT mechanism for 5G and beyond networks. The proposed mechanism allows for end-to-end monitoring using headers that incorporate information on latency and geolocation, thereby enabling steering policies without the need for SDN controller intervention.

SDNPS [95] is an SDN framework that utilizes P4 switches and implements an INT data packet format to support mMTC and URLLC slices. The INT data, which include queue occupancy, link throughput, and processing delay, are recorded in data packets forwarded by P4 switches and collected by an INT data collector module in the application plane.

TurboEPC [96] presents control plane offloading of a small amount of user state to MATs in P4 switches. These devices process a subset of signaling messages e.g., S1 release and service request.

SMARTHO [97] implements a smart handover procedure using programmable switches between 5G CU and DU. For this purpose, switches spoof the behavior of UEs and perform resource allocation in advance.

INCA [98,99] utilizes the SRv6 protocol for traffic identification and chaining. This solution is deployed using P4 software switches and Smart NICs, and it is capable of parsing SRH, TEID, and QoS ID.

GRED [100] presents a data placement and retrieval service for MEC to enhance routing path lengths and forwarding table sizes. This mechanism is implemented on a P4 device, which is not specified by the authors.

HDS [101] is a hybrid data sharing framework for hierarchical MEC. The data location service is divided into two parts: intra-region—based on a cuckoo summary—and inter-region, using a geographic MDT-based routing scheme. The authors did not provide any information on the specific programmable device utilized in their implementation.

Wu et al. [102] propose a coding scheme implemented in P4 switches for SDN-based 5G networks. This coding scheme performs arithmetic aggregation based on the residue number system (RNS) for multiple data packets generated by mMTC-IIoT devices.

Mallouhi et al. [103] present an in-network approach for adaptative beamforming towards the UE. In this approach, the UE periodically reports its location to a P4 hardware switch, which uses this information along with the base station information to compute the angle required to reconfigure the beam between the base station and the UE. The angle computations are approximated with MATs to improve computational efficiency.

Lotfimahyari et al. [104] propose an architecture that enables state replication between NFV instances by utilizing a custom publish-subscribe protocol that runs directly in P4 switches. This architecture is designed to reduce communication delays and processing overhead for NFVs.

### 4.4. Obtained Results Discussion

#### 4.4.1. Tunneling and Forwarding

The works in [46,49,55,59,64,68,69] report processing latencies of 1.23 μs, 200 μs, 25 μs, 0.458 μs, 6 μs, 5 μs, and 0.739 μs, respectively. The study in [56] reports a latency reduction of 77% compared to common software implementations, while [61] shows a drop of 3.75x compared to a host based UPF deployment using DPDK. Furthermore, [44] specifies an end-to-end latency of 50 μs, which is useful for checking compliance with the KPIs from [10].

Regarding throughput, experiments in [60,69,75] display 98.7 Gbps, 40 Gbps, and 2 Gbps, respectively, while [47] achieves approximately 100 Gbps. In [80], the obtained data rate is 402× higher than that of a pure UPF software design.

#### 4.4.2. Network Slicing

In [72,73], full isolation between slices is reported, with 512 users divided into 16 slices. The lowest priority slices experience a maximum delay of 2.5 ms, whereas the rest of the traffic shows less than 1 ms of delay. When applied to an industrial smart grid use case (i.e., uRLLC) [74], a round trip time (RTT) of 10.02 ms is obtained, with the authors reporting high reliability and only 0.3% packet loss when the system is under the highest level of stress. In [76,77], where another industrial application is presented, experiments show delays of 15 ms for digital twin app data, 30 ms for telemetry packets, and 50 ms for remote support, all of which use virtual queues with priority queuing. The study in [79] presents an e-health video transmission use case in which slices with a high QoS level have an average delay of less than 0.05 ms. Moreover, in [52], the authors include the committed information rate (CIR) ratio as an adjustment parameter in the hardware switches for the throughput of each supported vertical for three IoT scenarios. For instance, mMTC applications require a CIR of 100% to achieve a 100% throughput percentage without any packet loss. The study in [85] demonstrates that the proposed prototype can support up to 16,000 slices, while [80] reports that creating slices takes 400 ms, which is faster than slicing implementations that do not use programmable data planes. The network slicing framework in [86] shows an average delay seen at the end-host of 69.8 ms. The solution presented in [84] demonstrates that slices experiencing high traffic loads, defined as four or more flows within a slice, share the remaining bandwidth with a maximum difference of 4%.Furthermore, the bandwidth allocated to each flow within a slice varies only slightly, ranging from 0.54% to 8.23% of the total slice bandwidth.

Finally, in [81,82], network slicing in the fronthaul section of NG-RAN results in update routing entry latencies of less than 6 μs, with support for multipoint routing of 80 Gbps.

#### 4.4.3. Cybersecurity

In [88,89], the authors describe a firewall implementation that effectively blocks malicious traffic and can manage up to 1,024 flows with minimal added delay. Another study on firewalls, presented in [66], reports that security rules are updated within 10 ms with a confidence interval of 95%. Meanwhile, the DoS detection scheme described in [63] is evaluated based on the time that programmable switches take to extract features from packets, which is 110 μs. This is significantly faster than CPU-based solutions. In contrast, the DDoS mitigator outlined in [90] incurs a latency below 150 μs while managing 1 Gbps of traffic. In [91], a defense mechanism against cellular botnets restricts traffic to 40 requests per second. Additionally, the secret permutation implemented in [87] encodes and decodes IoT packet payloads at a line rate of 6.4 Tbps, which the authors claim to be the fastest rate reported in the literature.

#### 4.4.4. In-Band Telemetry

The study described in [93] demonstrates that the end-to-end round trip time (RTT) and dequeue time can be accurately measured using their implementation of INT. Another work in [94] reported that the INT load did not have a significant impact on end-to-end latency, as it was found to be below 1 ms. Moreover, INT-based traffic steering can reduce latency to less than 50 ms. The results of applying INT for network slicing in [95] show that uRLLC packet loss is reduced to almost zero compared to conventional software approaches. Meanwhile, for mMTC, it was observed that the average throughput per node is approximately equal to the packet generation rate, regardless of the number of nodes requesting connections. Finally, the implemented INT in [63] achieved an average latency reduction of around 6 ms, while the programmable switch was able to extract features in around 110 μs.

#### 4.4.5. Control Plane Offloading

According to the research in [96], throughput can improve by up to 102 times and latency can decrease by 98% when switch hardware stores state data. In [57] authors show that offloading 35.79% of PFCP messages to programmable devices results in a 57% increase in throughput compared to solutions without this feature. Additionally, the study conducted in [67] reports that network latency can be reduced by up to 40% compared to traditional multi-switch topology implementations that do not process NGAP packages.

#### 4.4.6. Other Uses

In [100], the proposed data placement approach is shown to achieve improved load balancing and below 30% path lengths when compared to alternative methods. While the data retrieval scheme introduced in [101] achieves a reduction of 50.21% in lookup paths and 92.75% in false positives compared to other state-of-the-art implementations. On the other hand, the publish-subscribe scheme proposed in [104] is found to result in more homogeneous traffic and lower state replication latency, although no latency measurements are included. The service function chaining (SFC) scheme introduced in [98,99] can classify and create a service sequence for traffic flows, which results in an increase of only 1% in flow completion time (FTC), a reasonable impact for 5G architectures. In [97], experimental results show that their proposed framework can reduce handover time by up to 18% and 25% for two and three sequences, respectively. According to the findings reported in [65], the average reduction in handover delay achieved by the proposed solution was 2.11 times compared to a host-based approach. Finally, the beamforming method presented in [103], which employs programmable devices, can calculate angle approximations within acceptable empirical error distributions for moderate user movement speeds of less than 90 km/h and control cycle times less than 100 ms. It is worth noting that the resource consumption of the method depends solely on the size of the TCAM table used in the programmable device.

### 4.5. Summary and Insights

The majority of studies investigating tunneling and forwarding functions are implemented within the UPF and edge-to-core section and have focused on lowering latency as well as enhancing throughput. The performance achieved in these studies is heavily influenced by the type of device used. Programmable hardware switches have been found to achieve higher throughput compared to FPGA boards and smart NIC implementations, whereas software switches demonstrate the lowest line rates. Nonetheless, hybrid designs aim to leverage the programmability features of smart NICs or software switches to offload more complex user-defined rules and achieve lower latencies.

Network slicing implementations are mostly deployed in the edge-to-core segment of systems, with use cases such as e-health, industrial IoT, and smart grids being evaluated to demonstrate the efficacy of programmable devices in achieving slice isolation and fairness. While a study has explored the deployment of network slicing in the NG-RAN section of the architecture, to date, no work has integrated this approach with transport network slicing to achieve an end-to-end programmable data plane solution.

Articles pertaining to cybersecurity focus on strategies for blocking malicious traffic and content permutation. However, this type of implementation is restricted due to the computational constraints of programmable devices.

INT applications reviewed in this study utilize programmable devices to append additional headers containing specific data from 5G and beyond networks. These data are subsequently collected by a control entity to optimize key parameters and processes, such as QoS, traffic steering, and network slicing. Notably, the reviewed deployments in this domain are currently limited to software switch implementations only.

Control plane offloading using programmable data planes is effective for handling simple signaling components that can take advantage of the high-speed properties of these devices.

On the other hand, the reviewed literature reveals other additional uses of programmable data planes in 5G and beyond architectures. These applications include data placement, data retrieval, data aggregation, publish–subscribe schemes, service function chaining (SFC), handover processes, and beamforming calculations.

While programmable data planes have demonstrated a broad range of applications for 5G and beyond architectures, there are general limitations that need to be addressed. For instance, programmable devices do not currently support complex functionalities such as automatic repeat request (ARQ) and ciphering. Furthermore, handling state transfer for numerous UEs can be challenging due to memory constraints. Finally, a lack of realistic experimental settings can make it difficult to obtain reliable results.

## 5. Open Points and Future Research Directions

As programmable data plane implementations become increasingly relevant in 5G and beyond architectures, there are some open points and potential research directions that remain to be explored. Although there have been major advancements in research, a number of obstacles still need to be overcome. Within this section of the review, attention will be drawn to several critical domains that need further exploration, including issues corresponding to scalability, performance, computational constraints, interoperability, and energy efficiency. By identifying those gaps and future research tracks, the aim is to encourage further research in this emerging field.

Scalability: One of the main challenges associated with using programmable data planes in the 5G and beyond context is ensuring scalability to accommodate the large number of users and sessions these networks must manage. However, limited storage capacity on programmable devices poses constraints on state transfer, prompting further exploration of potential solutions. A promising approach is to use simultaneous software and hardware programmable devices, with the former managing low-traffic levels and the latter handling high-traffic levels. The study in [49] is a noteworthy approach that involves transitioning user connections to the hardware device if they exhibit high rates of data transmission. Additional studies that explore diverse hybrid software/hardware models can be found in [50,58]. Another option for improving scalability is to incorporate a QoS scheduler within a programmable device to establish distinct queues, as proposed in [77]. Finally, a variation of multipath transport protocols such as QUIC could be employed to exploit available paths while minimizing data storage in intermediate nodes. A preliminary work on the compatibility of the aforementioned protocol with programmable devices is available in [105].Performance: Although programmable data planes have been effective in improving network throughput and reducing latency by offloading tunneling and forwarding of user plane data, challenges remain in optimizing the performance of 5G and beyond networks. Minimizing control plane intervention is desirable whenever possible. To this end, smart NICs offer the flexibility to manage complex user plane rules reducing control plane-user plane bottlenecks. The studies in [56,65] can be seen as notable examples of leveraging the capabilities of smart NICs for optimization. Additionally, developing simpler control protocols can aid in offloading tasks to the data plane. The initial concepts of this approach are showcased in [57]. Another path for improving system performance is to explore end-to-end network slicing solutions that incorporate programmable devices in the NG-RAN and edge-to-core sections of the network. A starting point is the work in [82], which showcases network slicing in the fronthaul section of the system architecture. More research is required to thoroughly explore the potential advantages offered by such solutions.Computational limitations: Programmable data plane devices have inherent constraints in performing complex computations, as they do not support floating-point arithmetic operations and can handle integer values only. As a result, network functionalities that rely on complex operations will not be supported. To tackle this issue, approximation algorithms can be utilized to trade off precision for improved network performance. An application-oriented implementation of an approximation scheme utilizing the longest prefix match for programmable devices calculations is demonstrated in [106]. Another possible solution is to assign non-supported computations to the control plane (e.g., general-purpose CPUs), which is capable of handling more complex operations, as in the scheme presented in [107]. Nevertheless, this methodology could potentially result in a rise in latency, which represents a prospective aspect to take into account in forthcoming research work.Interoperability: Ensuring compatibility between programmable data plane devices and existing network infrastructures, protocols, and services is crucial for successful deployment. To achieve this, interoperability mechanisms must be developed that facilitate the integration of programmable devices with legacy equipment. One potential approach is to create hybrid testbed environments that combine both programmable and non-programmable devices to assess the feasibility of interoperability mechanisms. Some examples of programmable data-plane-oriented testbeds are featured in [52,108,109]. Another promising option is to explore emerging technologies such as digital twins [110] for evaluating compatibility and identifying potential issues before deployment in real-world network architectures.Energy efficiency: With the growing concern for environmental sustainability, energy efficiency is becoming an essential aspect of network design for 5G and beyond technologies. Programmable data plane devices are typically implemented using power-hungry hardware such as FPGAs or ASICs. This can lead to high energy consumption and costs. However, none of the surveyed articles specifically address this vital aspect. To fill this gap in the literature, research is needed to evaluate the energetic impact of programmable devices operation and ultimately develop energy-efficient schemes that can reduce power consumption while maintaining high network performance. Although the study in [111] presents an implementation within a data center framework, its central focus lies in utilizing programmable devices to consolidate traffic and mitigate the energy consumption of servers and network components. This serves as a promising initial step that can be further extended to a telecommunication network setting.

## 6. Conclusions

This systematic review has conducted a comprehensive analysis of the latest advancements in programmable data plane applications for 5G and beyond architectures. The investigation has revealed that the majority of implementations are currently deployed in the edge-to-core segment of networks, utilizing hardware programmable devices like switches and smart NICs. Key applications observed include tunneling and forwarding, as well as network slicing. The results highlight the promising potential of programmable data planes in enhancing the performance, flexibility, and reliability of upcoming telecommunications networks. Finally, this study has also acknowledged the main challenges and potential directions for future research, highlighting the need for additional exploration in areas such as scalability, interoperability, performance enhancement, computational constraints, and energy conservation.

## Figures and Tables

**Figure 1 sensors-23-06955-f001:**
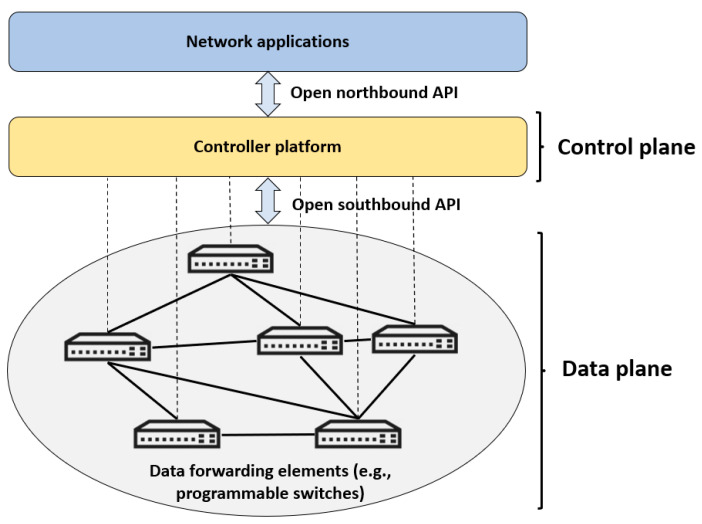
SDN general architecture [1].

**Figure 2 sensors-23-06955-f002:**
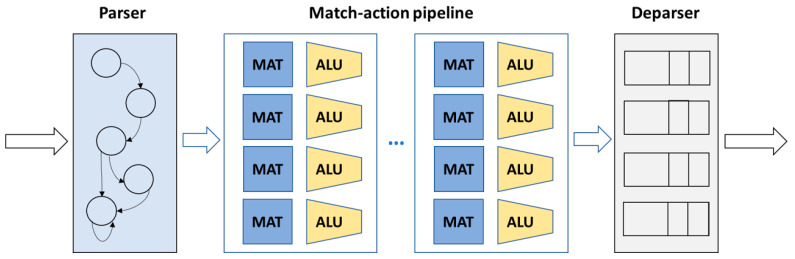
PISA programmable switch model.

**Figure 3 sensors-23-06955-f003:**
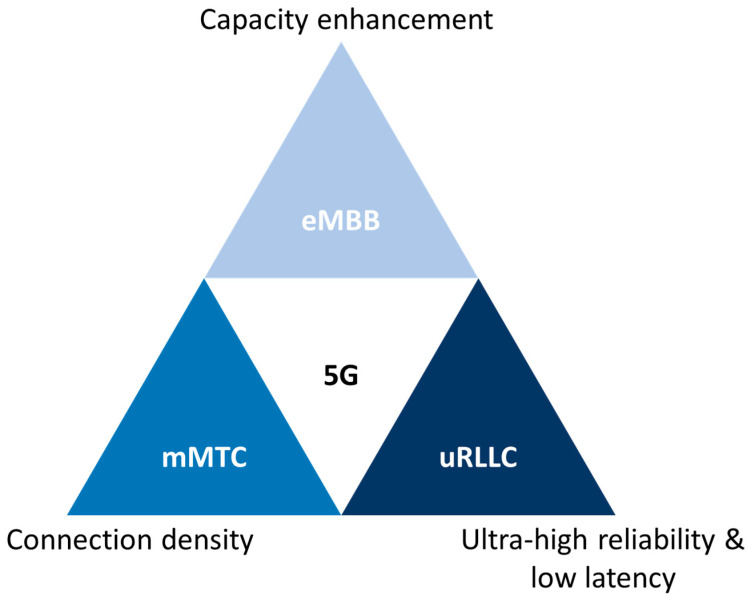
Categories for 5G applications [8].

**Figure 4 sensors-23-06955-f004:**
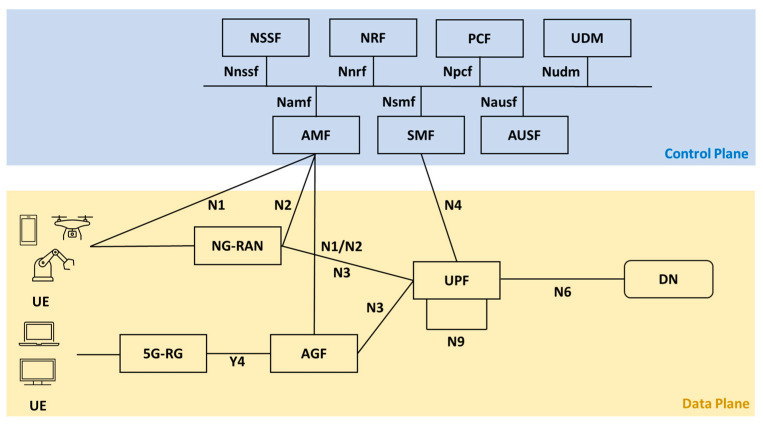
Simplified representation of the 5G system architecture.

**Figure 5 sensors-23-06955-f005:**
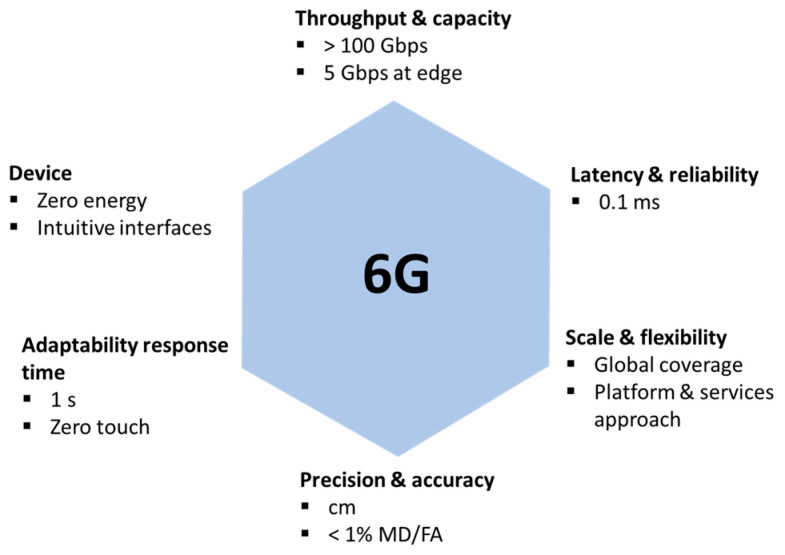
6G key requirements [11].

**Figure 6 sensors-23-06955-f006:**
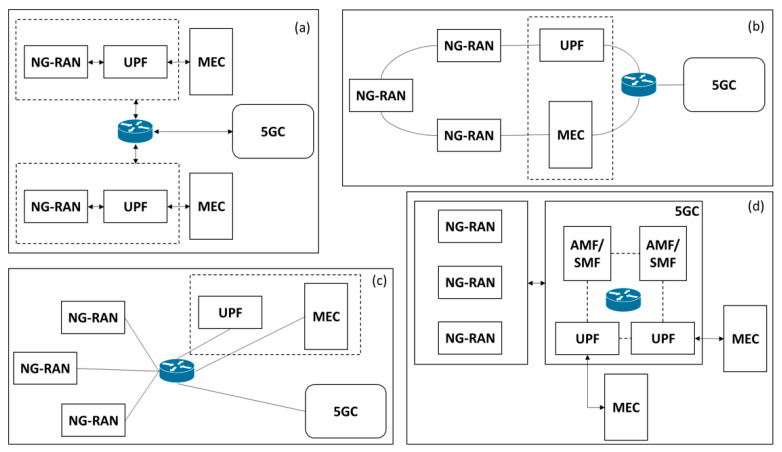
(**a**) NG-RAN and local UPF with MEC. (**b**) Transmission node and MEC with an optional local UPF. (**c**) Local UPF and MEC with network aggregation point. (**d**) 5GC functions with MEC [43].

**Figure 7 sensors-23-06955-f007:**
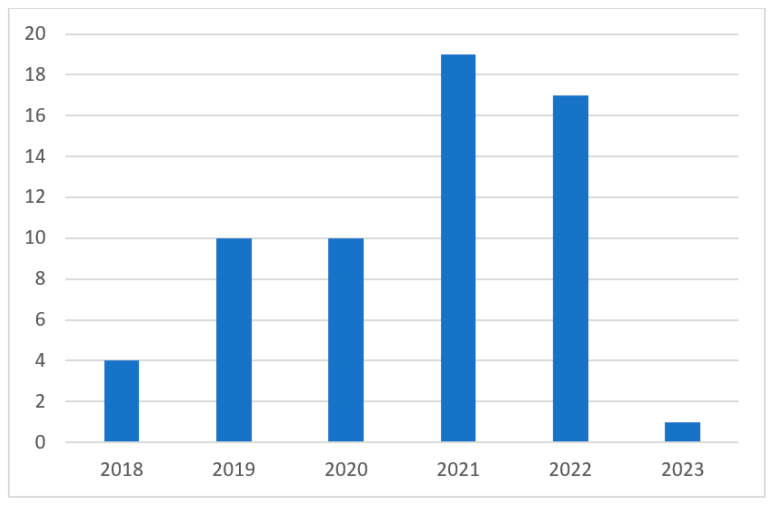
Distribution of reviewed articles per year.

**Figure 8 sensors-23-06955-f008:**
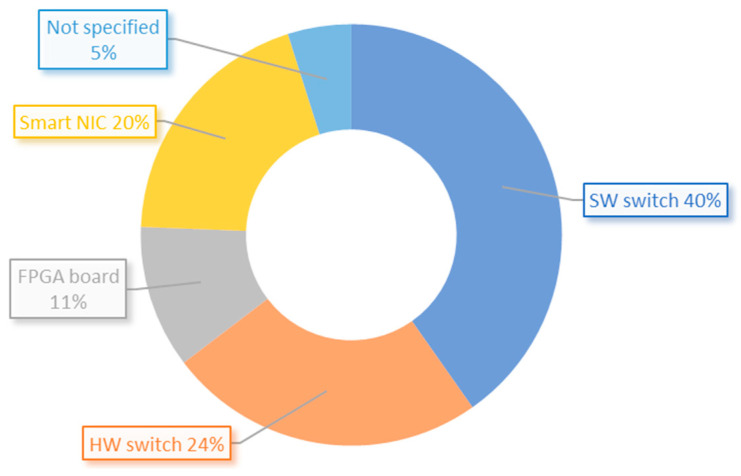
Distribution of implemented programmable devices from reviewed articles.

**Figure 9 sensors-23-06955-f009:**
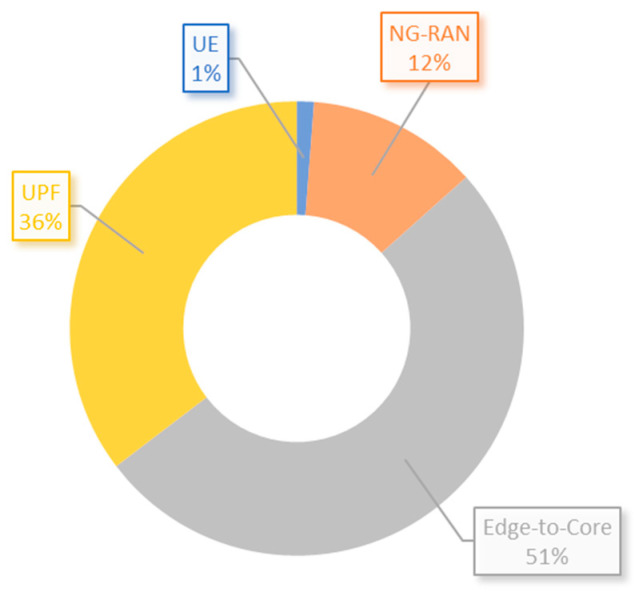
Architecture placement distribution of implementations from reviewed articles.

**Figure 10 sensors-23-06955-f010:**
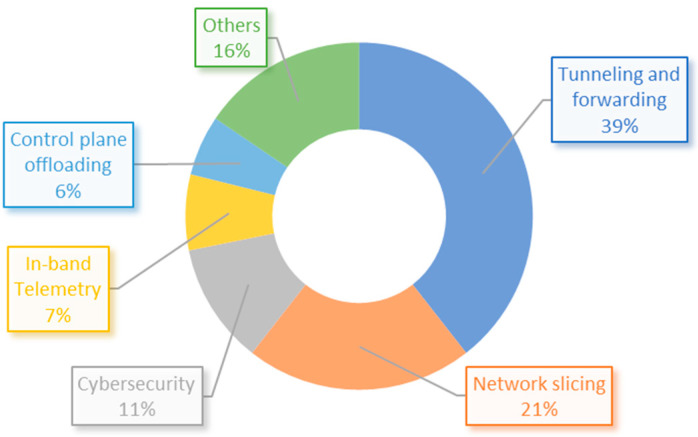
Use case distribution of implementations from reviewed articles.

**Table 1 sensors-23-06955-t001:** Minimum technical performance requirements of IMT-2020 [10].

KPI	Key Use Case	Values
Peak Data Rate	eMBB	DL: 20 Gbps, UL: 10 Gbps
Peak Spectral Efficiency	eMBB	DL: 30 bps/Hz, UL: 15 bps/Hz
User Experienced Data Rate	eMBB	DL: 100 Mbps, UL: 50 Mbps (Dense Urban)
5% User Spectral Efficiency	eMBB	DL: 0.3 bps/Hz, UL: 0.21 bps/Hz (Indoor Hotspot); DL: 0.225 bps/Hz, UL: 0.15 bps/Hz (Dense Urban); DL: 0.12 bps/Hz, UL: 0.045 bps/Hz (Rural)
Average Spectral Efficiency	eMBB	DL: 9 bps/Hz/TRxP, UL: 6.75 bps/Hz/TRxP (Indoor Hotspot); DL: 7.8 bps/Hz/TRxP, UL: 5.4 bps/Hz/TRxP (Dense Urban); DL: 3.3 bps/Hz/TRxP, UL: 1.6 bps/Hz/TRxP (Rural)
Area Traffic Capacity	eMBB	DL: 10 Mbps/m^2^ (Indoor Hotspot)
User Plane Latency	eMBB, uRLLC	4 ms for eMBB and 1 ms for uRLLC
Control Plane Latency	eMBB, uRLLC	20 ms for eMBB and uRLLC
Connection Density	mMTC	1,000,000 devices/km^2^
Energy Efficiency	eMBB	Capability to support high sleep ratio and long sleep duration to allow low energy consumption when there are no data (e.g., above 6 GHz)
Reliability	uRLLC	1–10^−5^ success probability of transmitting a layer 2 protocol data unit of 32 bytes within 1 ms in channel quality of coverage edge
Mobility	eMBB	Up to 500 km/h
Mobility Interruption Time	eMBB, uRLLC	0 ms
Bandwidth	eMBB	At least 100 MHz; up to 1 Gbps for operation in higher frequency bands

**Table 2 sensors-23-06955-t002:** Article classification based on system architecture and use case.

Use Case	Work	Architectural Placement				Device					Supported Technology
		UE	NG-RAN	Edge-to-Core	UPF	SW Switch	HW Switch	FPGA Board	Smart NIC	N/A	
Tunneling and forwarding	Aghdai et al. [44,45]			●					●		4G and 5G
	Shen et al. [46]				●			●			5G
	Lee et al. [47]				●		●				5G
	Singh et al. [48]				●		●				4G and 5G
	Singh et al. [49]		●		●		●		●		5G
	Vörös [50]		●				●				5G
	Ricart-Sanchez et al. [51]			●				●			5G
	Lin et al. [52]				●		●				5G
	NIKSS [53]				●		●				5G
	MacDavid et al. [54]				●	●	●				5G
	Alfredsson et al. [55]			●					●		5G
	Bose et al. [56]				●				●		5G
	AccelUPF [57]				●		●		●		5G
	CeUPF [58]				●		●				5G
	Rischke et al. [59]				●			●			5G
	Fernando et al. [60]		●	●		●					5G
	Jain et al. [61]				●				●		5G and beyond
	Gramaglia et al. [62]				●	●					5G and beyond
	BRAINE [63]				●	●					5G
	Kong et al. [64]			●			●				5G
	Synergy [65]				●				●		5G
	Velox [66,67]			●		●					5G
	Paolucci et al. [68]				●	●					5G
	Kundel et al. [69]				●		●				5G
	Kaloom 5G UPF [70]				●		●				4G and 5G
	Metaswitch Fusion Core [71]				●	●					4G and 5G
Network slicing	Ricart-Sanchez et al. [72,73,74]			●				●			5G
	Lin et al. [52]			●			●				5G
	Cunha et al. [75]			●		●					5G
	Chang et al. [76,77]			●		●					5G
	Chiu et al. [78]			●						●	5G
	Wang et al. [79]			●				●			5G
	FestNet [80]			●		●					5G and beyond
	FSA [81,82]		●				●				5G
	Yan et al. [83]			●					●		5G and beyond
	P4-TINS [84]			●			●				5G
	AHAB [85]			●			●				5G
	Turkovic et al. [86]			●		●	●				5G and beyond
Cybersecurity	Lin et al. [87]				●		●				5G
	Ricart-Sanchez et al. [88,89]			●				●			5G
	Paolucci et al. [90]			●		●					5G
	BRAINE [63]			●		●					5G
	Velox [66]			●		●					5G
	Wen et al. [91]				●	●					5G
	FrameRTP4 [92]			●		●					5G
In-band Telemetry	Paolucci et al. [90]			●		●					5G
	Dreibholz et al. [93]			●		●					4G and 5G
	Scano et al. [94]			●		●					5G and beyond
	SDNPS [95]			●		●					5G
	BRAINE [63]	●		●		●					5G
Control plane offloading	TurboEPC [96]		●		●	●			●		4G and 5G
	Bose et al. [56]				●				●		5G
	AccelUPF [57]				●		●		●		5G
	Velox [67]			●		●					5G
Handover	SMARTHO [97]		●			●					5G
	Aghdai et al. [45]		●						●		4G and 5G
	Synergy [65]				●				●		5G
Service function chaining	INCA [98,99]		●	●	●	●			●		5G
	FrameRTP4 [92]			●		●					5G
Data placement	GRED [100]			●						●	5G
Data retrieval	HDS [101]			●						●	5G
Data aggregation	Wu et al. [102]		●							●	5G
Beamforming calculations	Mallouhi et al. [103]		●	●			●				5G
Publish subscribe scheme	Lotfimahyari et al. [104]			●		●					5G

## Data Availability

Not applicable.

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
