# Peer review of "Programmable Data Plane Applications in 5G and Beyond Architectures: A Systematic Review"

_sensors, 2023, doi:10.3390/s23156955_

Round 1

Reviewer 1 Report

The paper is very good summary for starting the research in the field of 5G networks.

whereas, no clear contibution or results have been presented.

This paper is most likely review article not technical articel 

The manuscript has no models, results, analysis, or simulations.

Reviewer 2 Report

The article presents a systematic review of programmable data plane applications in 5G and beyond. While several articles in the literature cover this topic, the manuscript demonstrates a well-structured flow of content. However, I have the following comments and suggestions for improvement:

1) Section 3 contains a significant amount of general information that may dilute the article's focus. I suggest the authors consider including more specific details in this section.

2) It is recommended for the authors to include a dedicated section on related works. This will provide readers with a broader understanding of the existing research in the field and highlight the unique contribution of the current study.

3) While the review briefly discusses various programmable data plane applications, the authors should consider conducting a more critical evaluation of these applications. They can expand on the benefits, challenges, and potential impact within the context of 5G and beyond.

4) The authors should revise the layout of Table 2 to enhance readability and understanding for readers.

5) In Section 5, the authors should provide a more comprehensive review of recent advancements in research on challenges in programmable data plane frameworks. Specifically, they should focus on addressing critical issues highlighted in recent published literature.

6) What is the specific finding that the authors aim to convey in section 4.1?

7) The titles of Fig. 9 (a) and (b) do not accurately describe the content depicted in the figures.

The quality of English language in the article is generally acceptable.

Reviewer 3 Report

The paper is a useful contribution to the area of 5G. I believe it be scientifically sound. However, I have a few concerns that I would like to see addressed.

The Introduction to the paper needs to explain more clearly how SDN fits into 5G. The topic of SDN for 5G is not altogether an immediately obvious one. 5G is a cellular network where many of the ideas of 5G are already incorporated in SS7. I would very much like to see some discussion as to where SDN fits in to the already substantial signalling capabilities of 5G. What additional problem does it solve that are not already solvable within existing 5G signalling? 

The second issue is that the heart of the paper is Table 2 and Section 4.3 which goes through each of the publications summarising their contribution. Section 4.3 is very dull reading. I imagine most readers will skip it and go onto the next subsection where the contribution of each publication to the relevant use case is discussed. I am not going to insist on it but I would much prefer to see the content of 4.3 in a table. 

Section 4.4 is a good summary of the papers and their contributions to the 5G use cases.

Section 4.5 needs to be broken up into a few paragraphs. 

Section 5 is a good summary of the open research issues. 

A few typographical errors that should be addressed. Here is a list that of ones that I've identified:

186 "as a mean of" should be "as a means of"

193 "As it shown" should be "As shown"

484 Table heading should be with table.

766 "Summary and Found Insghts" to "Summary and Insights"

Round 2

Reviewer 1 Report

It is a review paper, but may be improved by inttroducing a survey on the obtained results 

Quality of English Language may be improved

Reviewer 2 Report

After thoroughly reviewing the authors' revisions in response to the first-round comments, I can observe an improvement in the manuscript's quality. The authors have addressed the raised concerns, and I recommend accepting the article for publication.

Minor editing of English language required.
